# Molecular basis for functional connectivity between the voltage sensor and the selectivity filter gate in *Shaker* K$^+$ channels

Carlos AZ Bassetto[1], João Luis Carvalho-de-Souza[1,2], Francisco Bezanilla[1,3,4]*

[1]Department of Biochemistry and Molecular Biology, The University of Chicago, Chicago, United States; [2]Department of Anesthesiology, University of Arizona, Tucson, United States; [3]Institute for Biophysical Dynamics, The University of Chicago, Chicago, United States; [4]Centro Interdisciplinario de Neurociencias, Facultad de Ciencias, Universidad de Valparaiso, Valparaiso, Chile

**Abstract** In *Shaker* K$^+$ channels, the S4-S5 linker couples the voltage sensor (VSD) and pore domain (PD). Another coupling mechanism is revealed using two W434F-containing channels: L361R:W434F and L366H:W434F. In L361R:W434F, W434F affects the L361R VSD seen as a shallower charge-voltage (Q-V) curve that crosses the conductance-voltage (G-V) curve. In L366H:W434F, L366H relieves the W434F effect converting a non-conductive channel in a conductive one. We report a chain of residues connecting the VSD (S4) to the selectivity filter (SF) in the PD of an adjacent subunit as the molecular basis for voltage sensor selectivity filter gate (VS-SF) coupling. Single alanine substitutions in this region (L409A, S411A, S412A, or F433A) are enough to disrupt the VS-SF coupling, shown by the absence of Q-V and G-V crossing in L361R:W434F mutant and by the lack of ionic conduction in the L366H:W434F mutant. This residue chain defines a new coupling between the VSD and the PD in voltage-gated channels.

*For correspondence:
fbezanilla@uchicago.edu

**Competing interests:** The authors declare that no competing interests exist.

## Introduction

Voltage-gated ion channels play fundamental roles in many physiological processes, such as generating the nerve impulse and regulating neuronal excitability, shaping the pacemaker in the heart, or controlling muscular contractility. Voltage-gated potassium channels (K$_V$) are members of the voltage-gated ion channels superfamily, responsible for the repolarization phase of the action potential, for maximal action potential firing frequency and for simply keeping the membrane potential negative in non-excitable cells. In general, K$_V$ channels are assembled as homotetramers. Each subunit monomer contains two functional domains: the voltage sensor domain (VSD), formed by transmembrane segments S1-S4, and the pore domain (PD), composed by transmembrane segments S5, P-loop (a reentrant region in the protein), and S6. The permeation pathway, located in the central axis of the channel, is formed by four PDs flanked by one VSD each.

Presently, there are two tetramerized architectures identified among K$_V$ channels: domain-swapped channels (K$_V$1.2 [*Long et al., 2005a*] and K$_V$7.1 [*Sun and MacKinnon, 2017*]) and non-domain-swapped channels (EAG1 [*Whicher and MacKinnon, 2016*], HERG [*Wang and MacKinnon, 2017*], HCN [*Lee and MacKinnon, 2017*], BK [*Tao et al., 2017*], K$_V$AP [*Tao and MacKinnon, 2019*], and KAT1 [*Clark et al., 2020*]). In domain-swapped channels, the VSD from one subunit is in close contact with the PD from a neighboring subunit, forming an extensive inter-subunit non-covalent interface. In non-domain-swapped channels, the VSD and the PD from the same subunit forms a similar non-covalent interface and the domains are connected by a shorter version of the S4-S5 linker.

According to a well-studied electromechanical coupling mechanism, henceforth canonical coupling for simplicity, S4 movements driven by changes in the membrane potential lead to changes in S4-S5 linker position, which results in the opening or closing of channel inner gate (bundle crossing), at the intracellular end of the permeation pathway (*Kalstrup and Blunck, 2018*; *Long et al., 2005a*; *Long et al., 2005b*; *Lu et al., 2002*). The VSD movements, generated by changes in membrane potential, produce gating currents and the charge-voltage (Q-V) curve can be estimated from the current time integral at different voltages. The maximal ionic conductance at each membrane potential serves to estimate the conductance-voltage (G-V) curve. Since VSD movements are the underlying trigger of voltage-dependent inner gate, the gating currents normally precede the pore opening, in voltage and time. Therefore, for a channel that opens as the voltage is made more positive, Q-V curves typically appear to the left of the G-V curves in the voltage axis.

However, the canonical coupling mechanism is not the only functional connection between the VSD and the PD. This can be inferred from an unexpected interaction between L361R, a mutation in S4, and W434F, a mutation in the PD (*Carvalho-de-Souza and Bezanilla, 2018*). W434F is a well-studied mutation, known to abrogate ionic $K^+$ currents by increasing the speed of C-type inactivation (*Perozo et al., 1993*; *Yang et al., 1997*) and it is also widely used to record gating currents in *Shaker* channels (*Bezanilla, 2018*). The L361R-W434F interaction was revealed by an unusual crossing between the Q-V curve (from L361R:W434F channels) and the G-V curve (from L361R channels) (*Carvalho-de-Souza and Bezanilla, 2018*). However, this did not occur when the Q-V curve was estimated from L361R channels, in the absence of W434F and after depleting $K^+$ ions. Therefore, the W434F mutation generated a channel with less voltage sensitivity and with a displaced Q-V curve. This suggests that the VSD and the PD are likely communicating by an alternative pathway, other than through the S4-S5 linker. Such an alternative pathway has been observed in *Shaker* (*Carvalho-de-Souza and Bezanilla, 2019*; *Conti et al., 2016*; *Fernández-Mariño et al., 2018*) and in $K_V7.1$ (*Hou et al., 2017*).

In *Shaker*, a domain-swapped channel (*Carvalho-de-Souza and Bezanilla, 2019*) showed that this alternative pathway operates independent of the canonical coupling. In fact, it was found to be mediated by the interface of contact between the VSD and the PD from different subunits. They classified this mechanism as noncanonical and demonstrated that it endows voltage dependence to the SF gate. They also hypothesized a series of amino acid residues as the molecular basis for this mechanism. These residues, shown in *Figure 1*, are arranged parallel to the plane of the membrane and span from the VSD (S4) to an adjacent PD (SF) (*Carvalho-de-Souza and Bezanilla, 2019*). At the VSD end, depending on the relative position of the S4, the series of residues starts with V363, L366, or V369. Next, comes L409, S411, and S412 in S5, F433, and W434 in the P-loop; at last, Y445 in the SF. From the $K_V1.2$ structure (*Chen et al., 2010*; *Long et al., 2005a*), it is possible to appreciate the close contact between these residues.

Here, we studied in detail the role of these residues in the coupling between the voltage sensor and the selectivity filter gate (VS-SF coupling). We assessed it by carrying out individual alanine substitutions at positions L409, S411, S412, and F433 on the L361R:W434F channel. These substitutions were chosen to decrease the van der Waals volume of a particular residue in an attempt to disrupt the functional connection between contiguous residues. All alanine substitutions restored the normal position of the Q-V curve with respect to the G-V curve. We used the same strategy on a mutant channel exhibiting evidence of a VS-SF coupling, the L366H:W434F channel. In this channel, L366H mutation in the VSD unexpectedly restored a fraction of the $K^+$ conduction abrogated by the W434F mutation. Interestingly, all alanine substitutions on L366H:W434F channels were sufficient to decrease the $K^+$ conduction to levels comparable to the W434F phenotype. This indicates that the VS-SF coupling has been disrupted. Together, these results demonstrate that the VS-SF coupling (1) comprises the proposed chain of residues, in domain-swapped channels; (2) is dependent on the volume of these residues; and (3) reveals that the VSD affects the dynamics of the SF gate and that the SF gate affects the movement of the VSD.

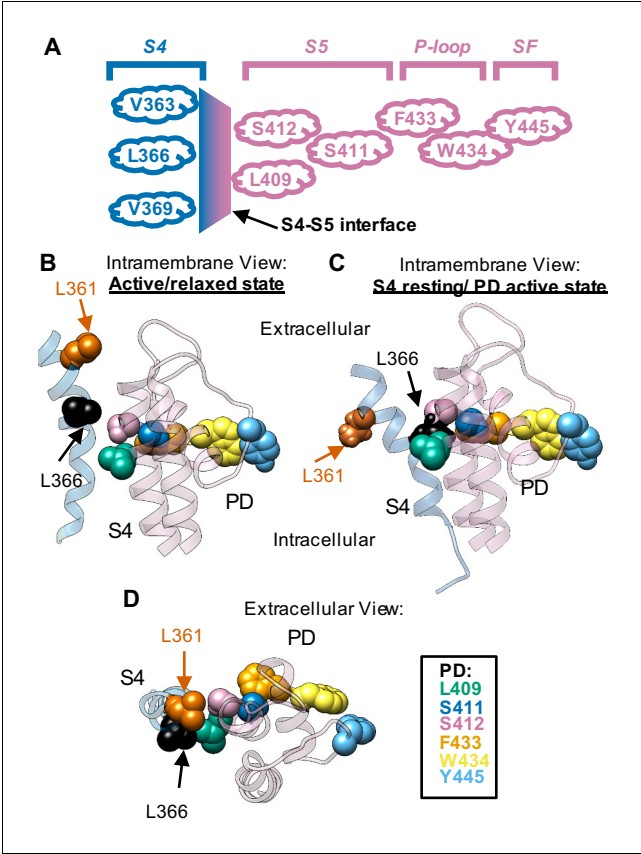

**Figure 1.** Proposed chain of residues involved in the voltage sensor selectivity filter gate (VS-SF) coupling. (**A**) Schematic representation of residues involved in the VS-SF coupling. Voltage sensor domain (VSD) is depicted in blue, whereas pore domain (PD) is depicted in light purple. (**B**) Intramembrane view of the active state. (**C**) Intramembrane view of S4 in resting state and PD in active state. (**D**) Extracellular view of the active state S4 segment (blue ribbon) and PD (light purple ribbon) are from an adjacent subunit, based on a computational 3D model of crystallographic structures of $K_V$1.2 (**Chen et al., 2010**) (Shaker-like channel, PDB: 3LUT). The model shows the VSD in the active/relaxed state for B and D. The S4 in the resting state is based on the consensus model (**Vargas et al., 2011**; **Vargas et al., 2012**). Residues proposed to be implicated in the VS-SF coupling mechanism are depicted by their van der Waals volumes (**Carvalho-de-Souza and Bezanilla, 2019**) (Shaker numbering): L361 (vermilion), L366 (black), L409 (green), S411 (blue), S412 (light purple), F433 (orange), W434 (yellow), and Y445 (sky blue). From the consensus model, it is possible to observe that L361 and L366 are practically in the same horizontal plane with the VS-SF coupling residues from the PD.

## Results

### Single alanine substitutions disrupt the VS-SF coupling in L361R:W434F channels

One at a time, we introduced the mutations L409A, S411A, S412A, and F433A in a *Shaker*-IR L361R: W434F background. *Shaker*-IR L361R:W434F channel studies indicate the mutation W434F actively changes VSD activation dynamics when bearing L361R mutation, evidenced by crossing Q-V and G-V curves. We found that Q-V and G-V curves do not cross each other when any of the alanine substitutions are performed in the presence of the W434F mutation (*Figure 2A-F*, *Table 1*). This indicates that the side chain volume in residues L409, S411, S412, and F433 is the base for the communication between the SF and the VSD. The Q-V curves from all mutants containing L361R were satisfactorily fitted with a two-state model equation and the best-fitted values for all curves used in this study are summarized in *Table 1*. A two-state model equation can satisfactory fit Q-V curves from sensors that appear to populate three or more conformational states during activation, but it does so by underestimating the total gating charges per sensor (*Bezanilla and Villalba-Galea,*

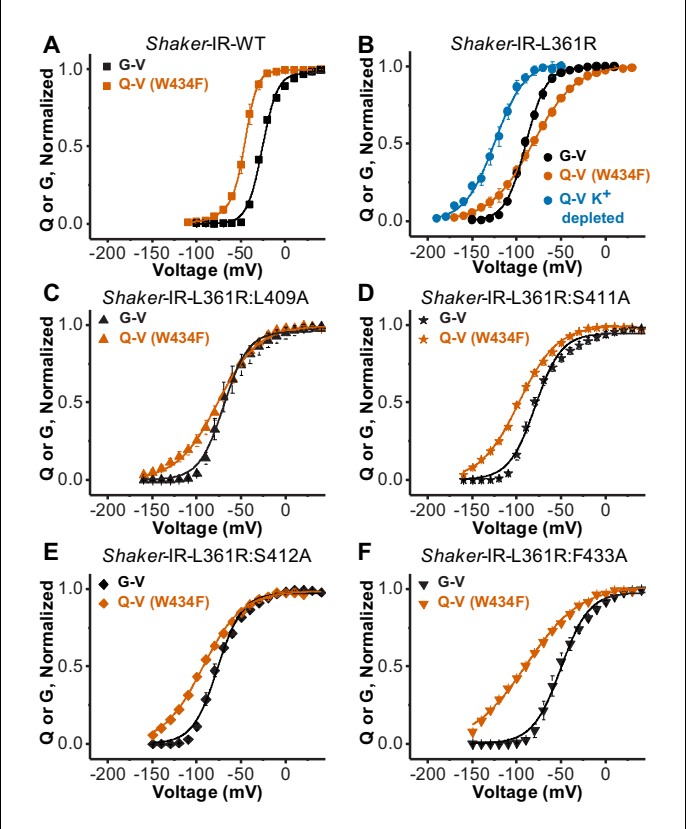

**Figure 2.** Disrupting the voltage sensor selectivity filter gate (VS-SF) coupling in Shaker-IR-L361R:W434F. (**A**) Conductance-voltage (G-V) (black) and charge-voltage (Q-V) (W434F-vermilion) curves from Shaker-IR-WT channels. Note the Q-V curve is displaced to more hyperpolarized potentials with respect to the G-V curve. (**B**) G-V (black) and the Q-V curves (W434F-vermilion) curves from Shaker-IR-L361R:W434F channels. Note that G-V crosses Q-V (W434F-vermilion), but not the Q-V when $K^+$ is depleted (blue). Adapted from **Carvalho-de-Souza and Bezanilla, 2018**. (**C**), (**D**), (**E**), and (**F**) are, respectively, G-V (black) and Q-V (W434F-vermilion) curves for Shaker-IR-L361R:L409A, Shaker-IR-L361R:S411A, Shaker-IR-L361R:S412A, and Shaker-IR-L361R:F433A channels. Note that when any of the residues L409, S411, S412, or F433 are mutated to Ala, the Q-V curve (W434F) does not cross the associated G-V curve, indicating the disruption of the VS-SF coupling. Continuous lines over Q-V and G-V curves are the best fit of **Equations 4** and 2 (two-state model), except the Q-V curve for WT channels that was fitted by **Equation 3** (three-state model). Best-fitted values are listed in **Table 1**. Plotted data are mean ± SEM (N = 4-8).

**2013**). Another possibility is that the W434F stabilizes the resting state of L361R VSD channel, affecting the VSD activation and/or deactivation, seen as a right shift in the Q-V curve. Altogether, it may be possible that channels bearing L361R mutations have more closed states than WT channels (an issue we did not pursue further in the present study).

## Effects by L366H mutation in S4 is consistent with a VS-SF coupling

The single amino acid mutation L366H added to a non-conductive Shaker-IR W434F channel background (L366H:W434F) yields a conductive one. Currents recorded from L366H:W434F channels appear to be highly expressed, as they clearly show gating currents followed by a slower rising current consistent with a $K^+$ conductance 1–2 times larger (**Figure 3a–c**). This is, however, a much smaller macroscopic conductance to gating currents ratio than in wild-type channels (typically two orders of magnitude or more). These data indicate that, although L366H:W434F channels are conductive, their unitary conductance and/or open probability are dramatically decreased.

Several features of the ionic currents carried by the L366H:W434F channels indicate that they represent $K^+$ moving through the permeation pathway of the PD and not through an eventual omega

**Table 1.** Best-fitted values for all the mutants studied.

| Mutant | Q-V | | | | | | | | G-V | | | | Inac-V | | | |
|---|---|---|---|---|---|---|---|---|---|---|---|---|---|---|---|---|
| | $V_0$ | | | $z_0$ | | | $V_1$ | | | $z_1$ | | $V_{1/2G}$ | | $z_G$ | | $V_{1/2Inac}$ | | $z_{Inac}$ | |

| Mutant | $V_0$ | $z_0$ | $V_1$ | $z_1$ | $V_{1/2G}$ | $z_G$ | $V_{1/2Inac}$ | $z_{Inac}$ |
|---|---|---|---|---|---|---|---|---|
| WT | | | | | −26.0 ± 0.6 | 2.9 ± 0.2 | −36 ± 0 | 4.6 ± 0.3 |
| W434F | −51.9 ± 1.9 | 1.7 ± 0.2 | −46.5 ± 1.1 | 3.4 ± 0.1 | | | | |
| L361R* | −124.3 ± 1.1 | 1.5 ± 0.1 | | | −89.7 ± 0.4 | 2.2 ± 0.1 | | |
| L361R:W434F | −81.5 ± 0.3 | 1.0 ± 0.0 | | | | | | |
| L361R:L409A | | | | | −69.4 ± 1.1 | 1.8 ± 0.1 | | |
| L361R:L409A:W434F | −76.2 ± 1.1 | 1.1 ± 0.0 | | | | | | |
| L361R:S411A | | | | | −79.2 ± 1.2 | 1.7 ± 0.1 | | |
| L361R:S411A:W434F | −97.4 ± 0.5 | 1.2 ± 0.0 | | | | | | |
| L361R:S412A | | | | | −76.4 ± 1.0 | 1.8 ± 0.1 | | |
| L361R:S412A:W434F | −93.2 ± 0.5 | 1.1 ± 0.1 | | | ± | ± | | |
| L361R:F433A | | | | | −50.1 ± 1.0 | 1.6 ± 0.1 | | |
| L361R:F433A:W434F | −89.3 ± 1.7 | 0.9 ± 0.1 | | | | | | |
| L366H | | | | | −52.5 ± 1.2 | 1.6 ± 0.1 | | |
| L366H:W434F | −87.2 ± 1.2 | 1.1 ± 0.1 | | | −52 ± 1 | 2.0 ± 0.1 | | |
| L366H:V478W | −89.8 ± 0.5 | 1.1 ± 0.1 | | | | | | |
| L409A | | | | | 0.2 ± 0.6 | 1.8 ± 0.1 | −20.6 ± 0.5 | 3.5 ± 0.2 |
| L409A:W434F | −62.4 ± 0.6 | 2.2 ± 0.1 | −36.3 ± 0.2 | 4.2 ± 0.1 | | | | |
| S411A | | | | | −11.9 ± 1.0 | 1.9 ± 0.1 | −28.7 ± 0.4 | 4.4 ± 0.3 |
| S411A:W434F | −67.2 ± 0.6 | 1.7 ± 0.1 | −44.4 ± 0.4 | 3.1 ± 0.1 | | | | |
| S412A | | | | | −20.0 ± 0.6 | 2.5 ± 0.1 | −28.0 ± 0.6 | 3.9 ± 0.3 |
| S412A:W434F | −60.9 ± 0.6 | 1.7 ± 0.1 | −37.2 ± 0.3 | 3.5 ± 0.1 | | | | |
| F433A | | | | | 6.6 ± 0.7 | 1.7 ± 0.1 | −14.6 ± 0.8 | 2.6 ± 0.2 |
| F433A:W434F | −44.0 ± 0.3 | 1.8 ± 0.1 | | | | | | |

*Q-V calculated by K+ depletion. Please note that L361R, L361R:W434F, L361R:S412A:W434F, L361R:F433A:W434F, L366H:W434F, L366H:V478W, L366H:S412A:W434F, and F433A:W434F, a two-state model Boltzmann equation, **Equation 4**, was used to fit the Q-V curves.

pore as in some *Shaker* mutants (**Tombola et al., 2005**) or being carried by protons as it happens with some other His mutations in the VSD (**Starace and Bezanilla, 2004**). First, the currents in the L366H:W434F mutant show a decay under sustained voltage-clamp depolarization reminiscent of C-type inactivation, suggesting this process has not been eliminated but rather mitigated (**Figure 3a,b**). Second, the currents were blocked by 1 mM of 4-aminopyridine (4-AP) (**Figure 3c,d**), a broad-spectrum K+ channel pore blocker (**Kirsch et al., 1993**). Third, the currents through L366H:W434F channels are K+-selective as revealed by their reversal potential in different K+ gradients (**Figure 3a,b,d**). Moreover, the maximal conductance in symmetrical K+ solutions was approximately threefold larger than the conductance in asymmetrical K+ solutions (**Figure 3e**). It has been reported that under symmetrical K+ concentrations (140 mM), the open probability (Po) of *Shaker*-IR-W434F is a 100-fold higher as compared to the absence of external K+ ($10^{-7}$ vs. $10^{-5}$) with no expected changes in the single channel conductance (**Yang et al., 1997**). In a different study, an increase in the number of channels available to open was observed in C-type inactivation enhanced mutants of *Shaker* (e.g. T449A, T449E, T449K) (**López-Barneo et al., 1993**). Therefore, we interpret the increased conductance in higher external K+ in L366H:W434F channels as a combination of the increase in the open probability and the increase of the number of channels that open under depolarization, which appears to be a common feature in C-type inactivation enhanced mutants of *Shaker*.

Moreover, G-V curves from L366H mutants in the absence and in the presence of W434F mutation have very similar voltage dependency and voltage sensitivity (**Figure 3f**, **Table 1**). This

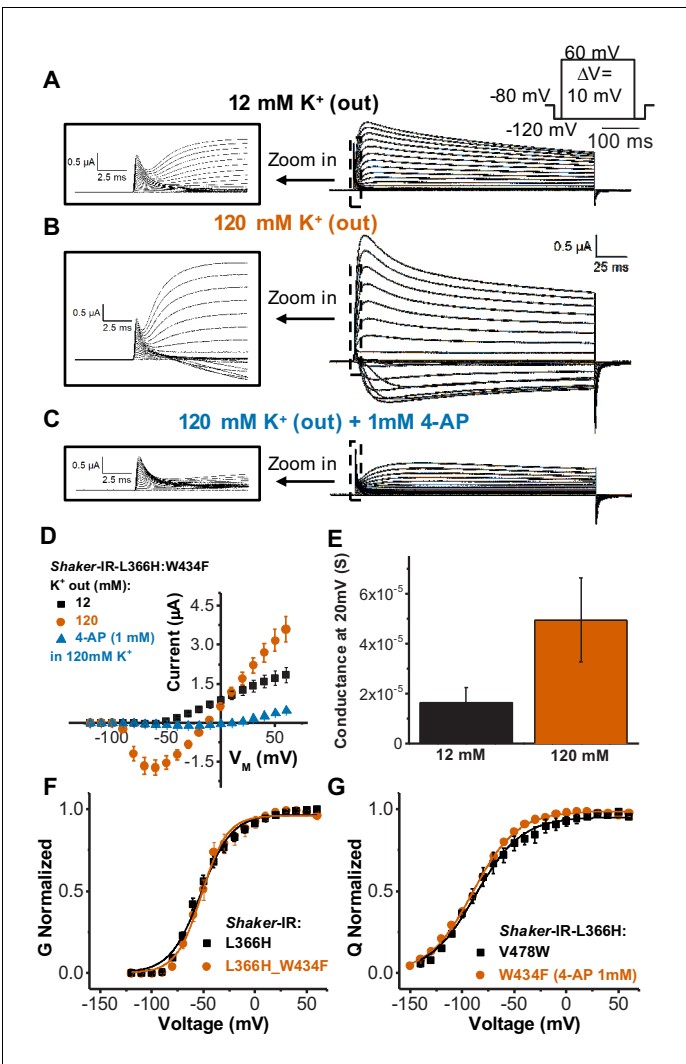

**Figure 3.** Single amino acid mutation in the voltage sensor partially relieves the inactivation in channels bearing W434F mutation. (**A**) Typical currents from Shaker-IR-L366H:W434F channels elicited in 120 mM internal and 12 mM external K$^+$ solution. (**B**) Currents elicited in symmetric 120 mM K$^+$ solutions. (**C**) Typical currents in the presence of 1 mM 4-aminopyridine (4-AP) in symmetric 120 mM K$^+$ solutions showing blockage of ionic currents (compared to B). Insets in A, B, and C are a time expansion to better visualize the gating currents. Voltage protocol used to record the currents is inset in A and currents presented in A, B, and C are from the same oocyte. (**D**) I-V relationships for mutant L366H:W434F in different ionic conditions, as indicated. (**E**) Conductance at 20 mV for L366H:W434F in 12 and 120 mM external K$^+$ concentration, and 120 mM internal K$^+$ concentration. We used 20 mV for comparison because the conductance-voltage relationship (G-V) at that voltage is already at its maxima. The conductance was calculated using the K$^+$ currents peak as described in 'Materials and methods'. (**F**) Normalized G-V curves for L366H and L366H:W434F. K$^+$ concentration used was 120 mM internal and 12 mM external. (**G**) Nearly identical charge-voltage (Q-V) curves measured with L366H:V478W (black squares) and L366H: W434F 1 mM 4-AP (vermilion circles). Continuous lines over G-V and Q-V curves are the best fittings of *Equations 2* and 4 (two-state model), respectively. Best-fitted values are listed in *Table 1*. Data shown as mean ± SEM (N=3).

observation further reinforces the notion that the PD of L366H:W434F channels is conductive and suggests that the canonical coupling underlying channel gating seems to be unaltered in this double mutant. Q-V curves measured in L366H channels with V478W mutation, that prevents inner gate (bundle crossing) opening (*Kitaguchi et al., 2004*) or W434F (after conductance blockade by 4-AP), are also practically identical (*Figure 3g – Table 1*). This implies that the W434F mutation does not affect VSD movements in the particular case of L366H-mutant VSD.

## Single alanine substitutions in the chained residues eliminate VS-SF coupling in L366H:W434F channels

If L366H and W434F are interacting and this interaction is carried out by the same coupling pathway as demonstrated for L361R and W434F case (*Figures 1* and *2*), then the ionic K$^+$ currents through L366H:W434F channels should be decreased to levels comparable to the W434F phenotype when a similar alanine mutation strategy is used. Consistently, mutating L409, S411, or S412 to alanine in L366H:W434F channels, the K$^+$ currents were greatly depressed leaving only gating currents to be observed (*Figure 4*). Once again, the magnitude of gating current recorded attests to the high level of protein expression. Peak of ionic currents from all triple mutants were remarkably small (to ~0.3 µA at 60 mV), indicating L409, S411, and S412 are also involved in the VS-SF coupling in L366H:W434F channels. Due to low expression levels, we were unable to record gating currents from L366H:F433A:W434F triple-mutant channels.

## Effects of single alanine mutations on C-type inactivation and VSD to pore conduction coupling

Even though alanine substitutions of amino acids within the connecting chain have been an effective strategy to delineate the VS-SF coupling, it is important to understand the functional consequences of such mutations in regular *Shaker*-IR-WT channels. We assessed the influence of those alanine mutations on the slow inactivation process and on Q-V and G-V curves (*Figure 5*). The weighted inactivation time constants ($\tau_{Inac}$) for WT, L409A, S411A, S412A, and F433A were assessed by fitting a double exponential equation to the decaying phase of K$^+$ currents elicited by long (19 s)

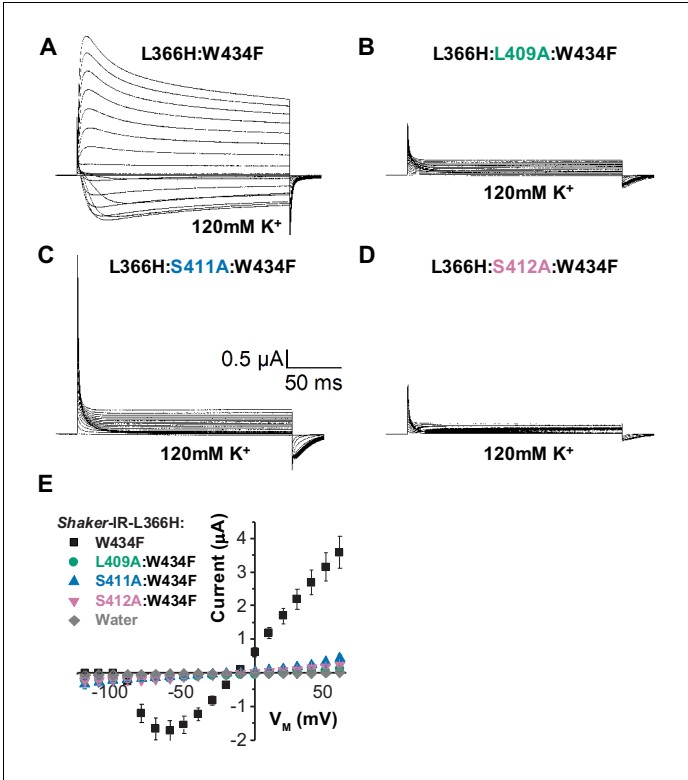

**Figure 4.** Disrupting the voltage sensor selectivity filter gate (VS-SF) coupling in Shaker-IR-L366H:W434F. (**A**) Typical currents from L366H:W434F channels elicited in symmetric 120 mM K$^+$ solutions as indicated. (**B**), (**C**), and (**D**) are currents also elicited in symmetric K$^+$ solutions for L409, S411, or S412 mutated to alanine in L366H:W434F channels, respectively. Voltage protocol used is the same shown in *Figure 3a*. Note that K$^+$ currents are not present in recording shown in B, C, and D, indicating the disruption of the VS-SF coupling. (**E**) I-V relationship taken from the peaks of K$^+$ currents show that when L409, S411, and S412 are mutated to Ala, the K$^+$ currents are dramatically diminished. The small remaining current is indistinguishable from leak. Data shown as mean ± SEM (N=4).

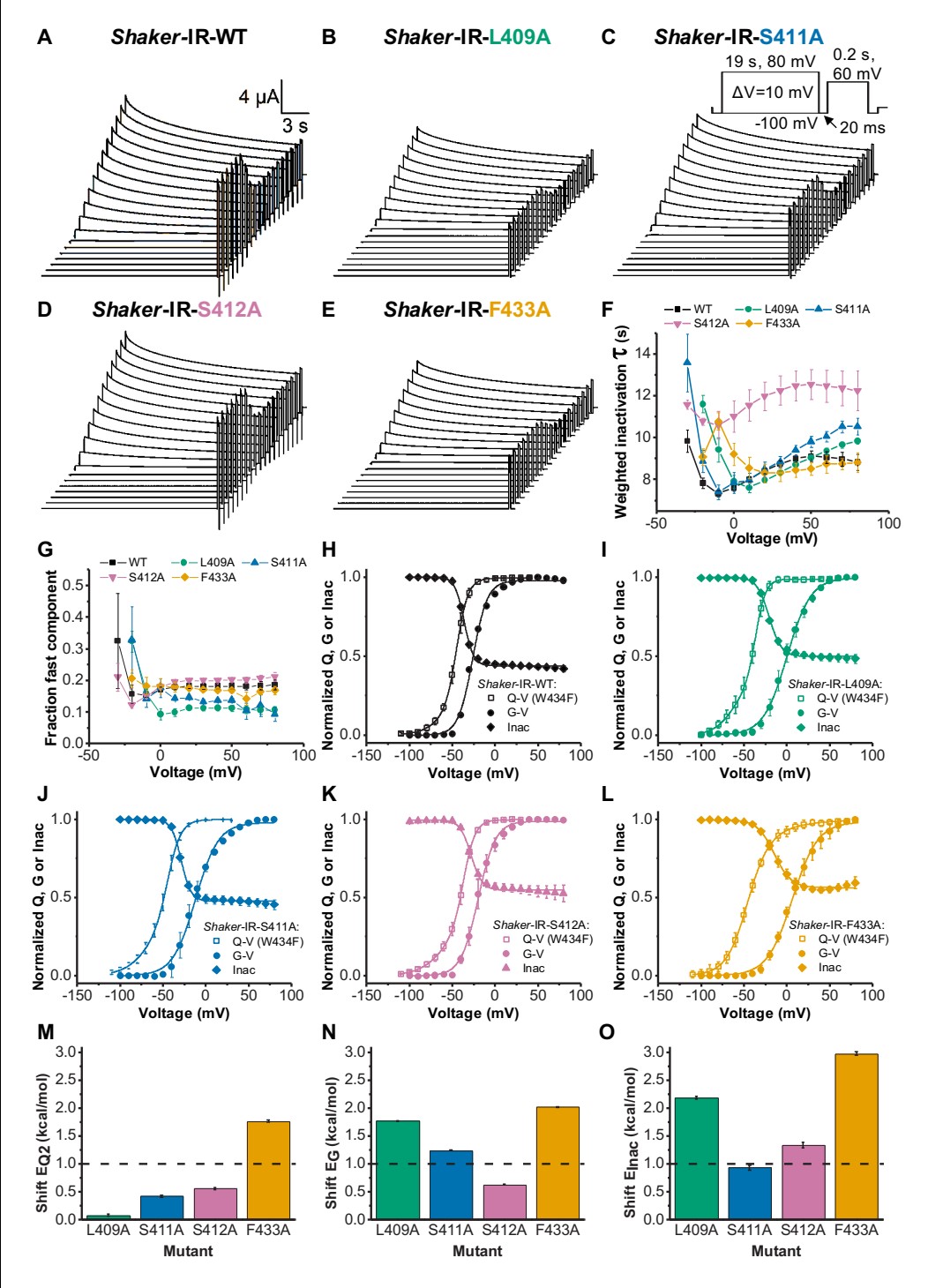

**Figure 5.** Voltage sensor domain (VSD) to pore opening coupling and C-type inactivation are affected by L409A, S411A, S412A, and F433A. **A, B, C, D, and E** are, respectively, typical K⁺ currents elicited by the voltage protocol inset in C (not to scale), for Shaker-IR-WT, Shaker-IR-L409A, Shaker-IR-S411A, Shaker-IR-S412A, and Shaker-IR-F433A channels. (**F**) Weighted inactivation time constants calculated from a double exponential fit to currents from WT, L409A, S411A, S412A, and F433A channels. (**G**) Fraction of the fast component of inactivation for channels tested as indicated in the graph. (**H, I, J, K, and L**) are, respectively, charge-voltage (Q-V) (open square), conductance-voltage (G-V) (solid circle), and Inac-voltage (Inac-V) curves (solid diamond) for WT (black), L409A (green), S411A (blue), S412A (light purple), and F433A (orange) channels. Currents used to build Inac-V curves were taken from the peak currents elicited by the 60 mV testing pulse after the 19 s inactivating period. **M, N, and**

*Figure 5 continued on next page*

*Figure 5 continued*

O are, respectively, shifts (from WT values) in the energy involved in the second component of Q-V, G-V, and Inact-V curves, as caused by L409A, S411A, S412A, and F433A mutations. Continuous lines over G-V, Q-V, and Inac-V curves are the best fits to *Equations 2, 3, and 7*, respectively. Q-V from F433A:W434F was fitted using *Equation 4*. The best-fitted values are listed in *Table 1*. Data shown as mean ± SEM for F to L; and mean ± SE for M to O (N=5-8).

depolarized voltage pulses (*Figure 5a–e*). While the $\tau_{Inac}$ was essentially identical for F433A and WT, it was considerably slower for L409A, S411A, and S412A (*Figure 5f*). The fraction of the fast component of the inactivation for L409A and S411A was smaller when compared to WT, but similar for S412A and F433A (*Figure 5g*).

Some of the alanine substitutions (L409A, S411A, and F433A) led to significant right shifts both in the G-V curve and in the available current after several different inactivating voltages pulses (Inac-V curve) (see *Figure 5h–l* and *Table 1*). The corresponding Q-V curves from those mutants, on the other hand, were not right-shifted, suggesting that those replaced residues exert a significant role in the VSD to pore conduction coupling in regular channels. We evaluated the change in the energy involved in both the activation and the inactivation processes, as induced by L409A, S411A, S412A, and F433A. To that end, we evaluated (see 'Materials and methods'): (1) the second component of the Q-V curves, taken from fitting a three-state model (*Equation 3*, see 'Materials and methods') ($E_{Q2}$, *Figure 5m*); (2) the conductance ($E_G$, *Figure 5n*); and (3) the C-type inactivation process ($E_{Inac}$, *Figure 5o*). We evaluated the second component of the charge movement because this transition is associated with the opening of the pore (*Ledwell and Aldrich, 1999*). A threshold of 1 kcal/mol, relative to WT, was set as a threshold for significance when considering the energetics of each process. Among all alanine substitutions, only F433A led to a significant change in the energy of the second transition of the Q-V curve (~1.7 kcal/mol – *Figure 5m*). For the energy associated to the conductance process, only mutation S412A did not lead to a significant change (*Figure 5n*). Whereas only mutation S411A did not lead to a significant change in the energy related to the inactivation process (*Figure 5o*).

Altogether, from these results we learned that, apart from F433A, alanine substitutions in the VS-SF coupling have little or no effect in the dynamics of WT VSD. Nevertheless, those same residues are crucially important for the channel VSD to pore conduction coupling. It involves both activation and inactivation processes even though the overall effects were small. Notably, the available current at very positive voltages was considerably increased when mutations S412A and F433A are present in the channels (*Figure 5h,k and l*), showing the importance of the residues to stabilize the inactivation conformation at the SF.

## Interaction between members of the chain does not depend on W434F mutation

We further explored the role of the VS-SF coupling by assessing a possible interaction between residues S411 and F433. The rationale for using those residues lays on their proximity in the protein, according to molecular models available. Another motivation to study the interaction between these two residues is that they have not been previously implicated with a noncanonical electromechanical coupling as shown for S412 (*Fernández-Mariño et al., 2018*). These two residues also do not affect the inactivation of the channels as W434F mutation does (*Perozo et al., 1993*). To record gating currents by a mechanism that does not involve the SF and W434F, we used the V478W mutation at the bundle crossing region. This mutation prevents ionic conduction (*Kitaguchi et al., 2004*; *Pau et al., 2017*) and does not affect the movement of the VSD (*Kitaguchi et al., 2004*). Therefore, we can assess the interaction between S411 and F433, in the absence of W434F. We used the thermodynamic mutant cycle and the generalized interaction-energy analysis to calculate this interaction (*Chowdhury et al., 2014*; *Chowdhury and Chanda, 2012*; *Figure 6*). This method uses the median voltage ($V_{Median}$) from the Q-V curves to estimate the free energy of activation. The $V_{Median}$ values calculated are shown in *Table 2*. The interaction between S411 and F433 is +2.00 ± 0.76 kcal/mol and is above the cutoff value of 1.8 kcal/mol, as previously established for this kind of analysis (*Chowdhury et al., 2014*; *Fernández-Mariño et al., 2018*). Therefore, this data supports that the connecting chain operates in the WT channels, independent of the presence of W434F mutation.

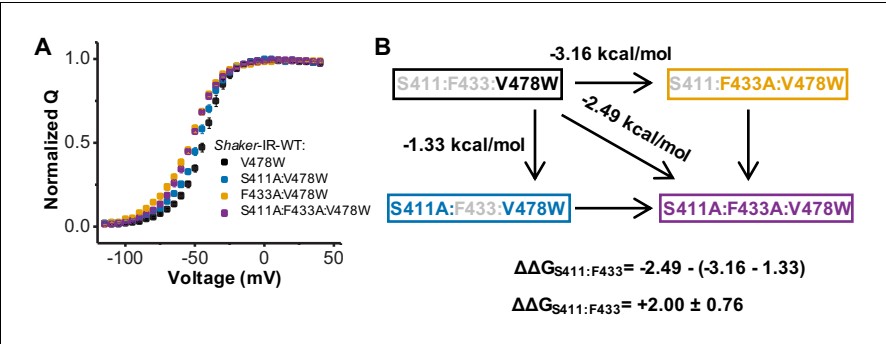

**Figure 6.** Interaction between residues S411 and F433 does not depend on W434F mutation. (**A**) Q-V for Shaker-IR-V478W (black), Shaker-IR-S411A:V478W (blue), Shaker-IR-F433A:V478W (orange), and Shaker-IR-S411A:F433A:V478W (purple). (**B**) Diagram showing the thermodynamic mutant cycle analysis used to estimate the interaction between residues S411 and F433. The energies were calculated using the generalized interaction-energy analysis (GIA). The $V_{Median}$ and energy calculated for each channel are shown in *Table 2*. Data shown as mean ± SEM (N=4-8).

## Discussion

Recently, several lines of evidence have pointed to the existence of different coupling mechanisms between the VSD and the PD in *Shaker* channels (*Fernández-Mariño et al., 2018*; *Conti et al., 2016*; *Carvalho-de-Souza and Bezanilla, 2019*; *Petitjean et al., 2015*). These mechanisms extend beyond the canonical electromechanical S4-S5 linker-based coupling. In this study, we demonstrate that a chain of amino acid residues is determinant for an alternative coupling in *Shaker*. Three residues in the S5 segment (L409, S411, S412) and two in the P-loop (F433, W434) are key for this functional VS-SF coupling. Crucially, the VSD and the PD involved in this chain are conformationally coupled through adjacent subunits. Yet, we must consider the fact that these aforementioned residues may not be the only ones involved in these processes, as it is possible that other paths may also exist (*Conti et al., 2016*; *Lee et al., 2009*).

We studied two functionally distinct abnormal W434F construct channels: L361R:W434F and L366H:W434F. In the first case, W434F affects the movement of L361R VSD seen as a shallower Q-V that crosses the G-V curve from channels not bearing W434F mutation. In the second case, L366H partially relieves the inactivation promoted by W434F, turning a non-conductive channel into a conductive one. The observation of these cases is quite remarkable because it shows the bidirectionality of the putative VS-SF coupling between the VSD and the PD. Irrespective of which abnormal W434F construct we used, single alanine substitutions to any of the residues of the chain are sufficient to disrupt or at least mitigate the abnormal phenotype of the W434F-bearing channels.

It is important to notice that all our observations showing an exacerbated VS-SF coupling were done in the presence of the W434F mutation. Under those conditions, one could argue that this mechanism might operate only after the channel populates the energy well equivalent to a C-type inactivated channel (the W434F phenotype). Our data suggest that the mechanism is in fact always present (*Figures 5* and *6*) and it is independent of the presence of W434F mutation; however, the W434F mutation makes it more conspicuous because this mutation significantly enhances C-type inactivation (*Yang et al., 1997*). Therefore, the roles of these chained residues on the overall VSD to pore opening coupling mechanism in regular Shaker channels are validated. Indeed, these results

**Table 2.** $V_{Median}$ and the free energy for the interacting residues S411 and F433.

| Mutant | $V_{Median}$ (mV) | | | G (kcal/mol) | | |
|---|---|---|---|---|---|---|
| V478W | −44.29 | ± | 1.24 | −13.90 | ± | 0.39 |
| S411A:V478W | −48.54 | ± | 1.13 | −15.24 | ± | 0.35 |
| F433A:V478W | −54.34 | ± | 1.38 | −17.06 | ± | 0.43 |
| S411A:F433A:V478W | −52.21 | ± | 1.11 | −16.39 | ± | 0.35 |

when combined demonstrate that there is a detectable coupling path between the VSD and the PD, as outlined in *Figure 1*.

Recently, a different study has shown that F244C, a mutation in S1 segment of *Shaker* channels, also produces a similar relief in the W434F-induced inactivation: the double mutant F244C:W434F also conducts K$^+$ and the current is blocked by 4-AP (*Petitjean et al., 2015*). F244, one of the residues forming the hydrophobic plug in *Shaker* channels (*Lacroix et al., 2014*), is in close proximity to residue S412 (*Chen et al., 2010*) from a PD of a neighboring subunit. Our data is consistent with these results, where mutations outside the PD, that is, L361R and L366H are able to interact with the W434F. Thus, under our proposal, the interaction between F244C and W434F would take place via the VS-SF coupling as well. Consistently, the F244C homolog in K$_V$1.1, F184C, causes episodic ataxia in human patients (*Browne et al., 1995*). Mutations in human K$_V$ channels at the S412 equivalent have neurological (in K$_V$1.1 [*Lee et al., 2004*] and K$_V$7.2 [*Dedek et al., 2003*]) and cardiac (in K$_V$7.1 [*Liu et al., 2002*]) consequences in humans.

In *Shaker* channel, the residue I470 is a key participant in the coupling between the activation gate, at the bundle crossing region, and C-type inactivation (*Peters et al., 2013*). It has been proposed that I470 interacts with T442, in the signature sequence of K$^+$-selective channels, TVGYG, at the SF (*Labro et al., 2018*). This mechanism constitutes an indirect pathway that couples the voltage sensor with the inner gate and the SF. Several studies have shown that mutations in the outer mouth of the pore, S5 segment and P-loop, affect C-type inactivation, highlighting the importance of these motifs for the inactivation process (*De Biasi et al., 1993*; *Larsson and Elinder, 2000*; *Molina et al., 1997*; *Ortega-Sáenz et al., 2000*; *Perozo et al., 1993*; *Yang et al., 1997*). These studies and our data demonstrate the existence of a direct interaction between the voltage sensor and the SF mediated by the mechanism proposed here.

The above discussion prompts two important questions: what is the contribution of the VS-SF coupling to the overall VSD to pore opening coupling? Furthermore, if the canonical mechanism is disrupted, can the channel be gated by the VS-SF coupling? Presently, we do not have data to support a definite answer for these questions, but it seems that the canonical coupling mechanism is the main mechanism whereby the channels' pore is gated by the VSD. Studies of channels with a stabilized open pore conformation may help evaluate the physiological role of the VS-SF coupling. Intersubunit metal bridges between residues V476C and H486 tend to stabilize the channels in the open conformation, displaying K$^+$ currents without apparent activation kinetics, as expected in the absence of a voltage-dependent gating process (*Holmgren et al., 1998*). A recent study, using T449A:V476C *Shaker* background and metal bridges, has concluded that the closure of the activation gate (where the canonical coupling mechanism operates) is essential for the recovery from C-type inactivation (*Szanto et al., 2020*). This suggests that the VS-SF coupling, by itself, is not sufficient to gate the channel's inner gate and, consequently, the channel. Furthermore, mutations in the bundle crossing region have shown that the pore stays open and does not close even when very negative potentials are applied (*Kitaguchi et al., 2004*). Regardless, an experimental strategy is yet to be developed to determine the role of VS-SF coupling in the absence of the canonical one.

It is reasonable to believe that the VS-SF coupling cannot, by itself, transfer enough energy to the PD in order to gate the channel, pointing to the notion that it is an accessory or modulatory gating mechanism. Indeed, this alternative mechanism might be more relevant in non-swapped domain channels such as from ether-a-go-go (EAG) K$^+$ channels family (EAG and hERG), which can open even without covalent interaction between the VSD and the PD (without the classical canonical coupling) (*Lörinczi et al., 2015*). This result suggests that the multiple non-covalent interactions between amino acid residues from the VSD and the PD can endow voltage dependence to the open probability of the PD. Furthermore, in hERG a mutation at W568 has been previously proposed to be involved in a pathway that couples activation and inactivation and when mutated to Leu (W568L), the channel does not exhibit C-type inactivation (*Ferrer et al., 2011*). Interestingly, residue W568 in hERG is homolog to residue S412 in *Shaker* demonstrating the physiological relevance of such alternative coupling mechanism in other members of K$^+$ channels family.

In summary, we have shown that residues L409, S411, S412, F433, and W434F comprise the molecular chain of residues that directly connects the voltage sensor to the SF gate, in *Shaker* K$^+$ channels. Our data also corroborated and clarified the idea of an alternative coupling mechanism. It is also possible that this mechanism is not only important in domain-swapped and non-swapped K$^+$

channels, but also may play a role in other members of the voltage-gated ion channels superfamily, for example, $Na_V$ channels.

# Materials and methods

### Key resources table

| Reagent type (species) or resource | Designation | Source or reference | Identifiers | Additional information |
|---|---|---|---|---|
| Gene *Drosophila melanogaster* | *Shaker* zH4 $K^+$ channel with Inactivation removed | *Hoshi et al., 1990* doi:10.1126/science.2122519 | | |
| Biological sample (*Xenopus laevis*, female) | Oocytes | Nasco | #LM00531 | Protocol #71475 (IACUC) |
| Chemical compound, drug | 4-Aminopyridine | Sigma-Aldrich | 275875–1G | 1 mM in dissolved in external solution |

### Site-directed mutagenesis

*Shaker* zH4 $K^+$ channel with fast inactivation removed (*Hoshi et al., 1990*), $\Delta 6$–46, and cloned into pBSTA vector was used. Mutations were performed using Quick-change II technology (Stratagene, La Jolla, CA), together with custom primers from Integrated DNA Technologies (Integrated DNA Technologies, Inc, Coralville, IA). *Shaker* cDNA and its mutants were sequenced, linearized by endonuclease NotI (New England Biolabs, Ipswich, MA), and cleaned up with a NucleoSpin Gel and PCR Clean-up kit (Macherey-Nagel, Bethlehem, PA). In vitro transcription kits were used to transcribe cDNA and generate cRNA (T7 RNA expression kit; Ambion Invitrogen, Thermo Fisher Scientific, Waltham, MA).

### Oocytes preparation

Oocytes were harvested from *Xenopus laevis* in accordance with experimental protocols #71475 approved by the University of Chicago Institutional Animal Care and Use Committee (IACUC). Following the follicular membrane digestion by collagenase, oocytes were incubated in standard oocytes solution (SOS) containing in mM: 96 NaCl, 2 KCl, 1.8 $CaCl_2$, 1 $MgCl_2$, 0.1 EDTA, 10 HEPES, and pH set to 7.4 with NaOH. SOS was supplemented with 50 µg/ml gentamycin to avoid contamination during incubation. After 6-24 hr of harvesting, defolliculated oocytes stage V-VI were injected with cRNA (5-100 ng diluted in 50 nl of RNAse free water) and incubated for 1-3 days at 12°C or 18°C prior to recording.

### Electrophysiological recordings

Ionic and gating currents were recorded using cut-open oocyte voltage-clamp method (*Stefani and Bezanilla, 1998*). Voltage-measuring pipettes were pulled using a horizontal puller (P-87 Model, Sutter Instruments, Novato, CA) to a tip resistance ranging from 0.2 to 0.5 MΩ. Currents were acquired by a setup comprising a Dagan CA-1B amplifier (Dagan, Minneapolis, MN) with a built-in low-pass four-pole Bessel filter for a cutoff frequency of 20–50 kHz. Using a 16-bit A/D converter (USB-1604, Measurement Computing, Norton, MA) for acquisition and controlled by an in-house software (GPatch64MC), data were sampled at 1 MHz, digitally filtered at Nyquist frequency and decimated for a storage acquisition rate of 50–200 kHz. Capacitive transient currents were compensated using a dedicated circuit. The voltage clamp was also controlled using the same in-house software and the 16-bit D/A converter of the same USB-1604 device.

For ionic current measurements, the external solution was composed by (mM): K-methanesulfonic acid (MES) 12, Ca-MES 2, HEPES 10, EDTA 0.1, *N*-methyl-D-glucamine (NMDG)-MES 108, pH 7.4. For 120 mM external $K^+$ solution, we replace NMDG with $K^+$. The cut-open oocyte internal solution was composed by (mM): K-MES 120, EGTA 2, HEPES 10, pH 7.4. For gating current recordings, we replaced $K^+$ by NMDG in both internal and external solutions. A conditioning hyperpolarized pre-pulse was used with different voltages to ensure that most of the channels were closed prior to the depolarizing pulses. The holding potential was set to $-80$ or $-100$ mV according to the mutant.

Recordings were performed at room temperature (~17–18°C). All chemicals used were purchased from Sigma-Aldrich (St. Louis, MO).

## Data analysis

The peaks of macroscopic K$^+$ currents were converted into conductance ($G$), using the equation below:

$$G = \frac{I_m}{V_m - \frac{RT}{F}\ln\left(\frac{[K^+]_{out}}{[K^+]_{in}}\right)} \tag{1}$$

where $I_m$ is the K$^+$ current, $V_m$ is the membrane voltage, $R$ is the gas constant, $T$ is the temperature in Kelvin, $F$ is the Faraday constant. $[K]_{in}$ and $[K]_{out}$ are the intracellular and extracellular K$^+$ concentrations, respectively.

Each $G$ value was normalized by the maximum $G$ of the experiment for averaging among experiments and plotting against $V_m$ in order to obtain a G-V curve. The G-V curves were fitted using a two-state model with the following equation:

$$G(V_m) = \frac{1}{1 + \exp\left(\frac{z_G F}{RT}\left(V_{1/2} - V_m\right)\right)}, \tag{2}$$

where $z_G$ is the apparent charge of the transition expressed in units of elementary charge ($e_0$) and $V_{1/2}$ is the voltage for 50% of the maximal conductance.

The charge ($Q$) was calculated from the integral in time of the gating currents. They were normalized, averaged, and plotted against $V_m$ to generate a Q-V curve. The Q-V curves were fitted using a three-state model (*Lacroix et al., 2012*):

$$Q(V_m) = N\,\frac{z_1 + z_0\left(1 + \exp\left(\frac{z_1 F}{RT}(V_1 - V_m)\right)\right)}{1 + \exp\left(\frac{z_1 F}{RT}(V_1 - V_m)\right)\left(1 + \exp\left(\frac{z_0 F}{RT}(V_0 - V_m)\right)\right)}. \tag{3}$$

where $N$ is proportional to the number of VSDs, $z_0$ and $z_1$ are apparent charges for the first and the second lumped sequential transitions of the VSD movement, respectively, $V_0$ and $V_1$ are the membrane voltages that split the VSD into two states equally during the first and second transitions, respectively.

In some cases, a two-state model was used to best fit the Q-V curves, as shown below:

$$Q(V_m) = \frac{1}{1 + \exp\left(\frac{z_q F}{RT}\left(V_{1/2} - V_m\right)\right)}, \tag{4}$$

where $z_G$ is the apparent charge of the transition expressed in units of elementary charge ($e_0$) and $V_{1/2}$ is the voltage for 50% of the maximal charge.

Inactivation time constants were taken from the decaying phase of K$^+$ currents after sustained activation by depolarizing voltage pulses. They were fitted using a double exponential shown below:

$$y(t) = y_0 - A_1 \exp\left(-\frac{t}{\tau_1}\right) - A_2 \exp\left(-\frac{t}{\tau_2}\right). \tag{5}$$

where $A_1$ and $A_2$ are the amplitudes from the first and the second exponential, respectively, $\tau_1$ and $\tau_2$ are the time constants for the first and second exponentials, respectively, and $y_0$ is a baseline adjustment.

For simplicity and analysis purposes, we considered a weighted time constant, calculated by the following equation:

$$\tau_w = \frac{A_1\,\tau_1 + A_2\,\tau_2}{A_1 + A_2} \tag{6}$$

where $\tau_w$ is the weighted time constant.

In order to build Inac-voltage (Inac-V) curves for each mutant, the peak K$^+$ currents from five to eight ooctyes were taken from the test pulse as shown in *Figure 5A*. They were normalized by their

maxima, averaged and plotted against $V_m$. A two-state model was used to fit the Inac-V curves by the equation below:

$$Inac(V_m) = A_2 + \frac{A_1 - A_2}{1 + \exp\left(\frac{z_{Inac}F}{RT}\left(V_{1/2Inac} - V_m\right)\right)}, \tag{7}$$

where $z_{Inac}$ is the apparent charge of the transition expressed in units of elementary charge ($e_0$) and $V_{1/2Inac}$ is the voltage dependence of the inactivation process. A₁ and A₂ are the minimum and the maximum value from the data, respectively.

We calculated the free energy ($G_{process}$) of the studied processes (second component of charge movement [$Q2$], conductance activation [$G$], and inactivation [$Inac$]) by using $V_{1/2Process}$ and $z_{Process}$ from the Q-V, G-V, and Inac curves. We used the following equation to calculate the free energies:

$$G_{Process} = z_{Process}FV_{Process}. \tag{8}$$

where $G_{process}$ is the free energy involved in the process, $z_{process}$ is the apparent charge involved in the process, $F$ is the Faraday constant, and $V_{process}$ is the voltage necessary to complete half of the total process (the $V_{1/2}$).

The shift $E_{Process}$ of the studied process was calculated using the following equation:

$$Shift\ E_{Process} = G_{Mutant} - G_{WT}. \tag{9}$$

The standard error (SE) associated to the energies were calculated using the following equation:

$$\delta Shift\,E_{Process} = F\sqrt{\left(\delta z_{Process}V_{1/2Process}\right)^2 + \left(\delta V_{1/2Process}z_{Process}\right)^2}. \tag{10}$$

To test the interaction between two residues, we used the thermodynamic mutant cycle, calculating the free energy using the generalized interaction-energy analysis that uses the $V_{Median}$ from the Q–V curves (*Chowdhury and Chanda, 2012*; *Chowdhury et al., 2014*). The free energy of activation ($G$) for a channel is calculated using the equation:

$$G = zFV_{Median} \tag{11}$$

The uncertainty associated in $G$ ($\delta G$) is:

$$(\delta G) = zF\delta V_{Median} \tag{12}$$

The number of charges ($z$) was considered the same for all channels: 13.6 elementary charges (*Schoppa et al., 1992*; *Aggarwal and MacKinnon, 1996*; *Seoh et al., 1996*).

The interaction free energy between residues S411 and F433 is:

$$\begin{aligned}\Delta\Delta G_{S411:F433} = &\ (G_{S411A:F433A:V478W} - G_{V478W}) - \\ &\ [(G_{S411A:V478W} - G_{V478W}) + (G_{F433A:V478W} - G_{V478W})]\end{aligned} \tag{13}$$

The subscripts refer to channel used to calculate the free energy. The uncertainty associated to $G_{S411:F433}(\delta G_{S411:F433})$ is given by:

$$\delta\Delta\Delta G_{S411:F433} = \sqrt{\delta V_{MedianV478W}^2 + \delta V_{MedianS411A:V478W}^2 + \delta V_{MedianF433A:V478W}^2 + \delta V_{MedianS411A:F433A:V478W}^2}. \tag{14}$$

$\Delta V_{MedianV478W}$, $\delta V_{MedianS411A}$, $\delta V_{MedianF433A}$, and $\delta V_{MedianS411A:F433A}$ are, respectively, the standard error of the mean associated with the measurement of $V_{Median}$ for V478W, S411A:V478W, F433A:V478W, and S411A:F433A:V478W channels.

Matlab (The MathWorks, Inc, Natick, MA) and Origin9.0 (Origin Lab Corporation, Northampton, MA) were used for calculating G-Vs, Q-Vs, plotting and fitting the data. Data is shown as mean ± SEM.

## Acknowledgements

We are deeply in debt with Drs Miguel Holmgren and Eduardo Perozo, and with Fraol Galan for their careful reading, comments, and suggestions on the manuscript. We thank Li Tang for helping with molecular biology. Supported by NIH grant R01-GM030376.

## Additional information

### Funding

| Funder | Grant reference number | Author |
| --- | --- | --- |
| National Institutes of Health | R01-GM030376 | Francisco Bezanilla |

The funders had no role in study design, data collection and interpretation, or the decision to submit the work for publication.

### Author contributions

Carlos AZ Bassetto, Conceptualization, Data curation, Software, Formal analysis, Validation, Investigation, Visualization, Methodology, Writing - original draft; João Luis Carvalho-de-Souza, Conceptualization, Formal analysis, Validation, Investigation, Visualization, Writing - original draft, Writing - review and editing; Francisco Bezanilla, Conceptualization, Resources, Formal analysis, Supervision, Funding acquisition, Validation, Investigation, Visualization, Methodology, Project administration, Writing - review and editing

### Author ORCIDs

Carlos AZ Bassetto (iD) https://orcid.org/0000-0002-7012-5699
João Luis Carvalho-de-Souza (iD) https://orcid.org/0000-0003-3231-0436
Francisco Bezanilla (iD) https://orcid.org/0000-0002-6663-7931

### Ethics

Animal experimentation: This study was performed in strict accordance with the recommendations in the Guide for the Care and Use of Laboratory Animals of the National Institutes of Health. All of the animals were handled according to approved Institution of Animal Care and Use Committee (IACUC) protocols (#71745) of the University of Chicago.

### Decision letter and Author response

Decision letter https://doi.org/10.7554/eLife.63077.sa1
Author response https://doi.org/10.7554/eLife.63077.sa2

## Additional files

### Supplementary files

• Transparent reporting form

### Data availability

All data generated or analysed during this study are included in the manuscript and supporting files.

The following previously published dataset was used:

| Author(s) | Year | Dataset title | Dataset URL | Database and Identifier |
| --- | --- | --- | --- | --- |
| Carvalho-de-Souza JL, Bezanilla F | 2018 | Nonsensing residues in S3-S4 linker's C terminus affect the voltage sensor set point in K+ channels. | https://rupress.org/jgp/article/150/2/307/43720/Nonsensing-residues-in-S3-S4-linker-s-C-terminus | Fig 2B, JGP |

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
