## [Decision Letter]

**Acceptance summary:**

This study by Basseto at al. examines the mechanism of voltage-dependent activation and inactivation in a prototypical voltage-gated ion channel. They identify a network of interacting residues that are critical for coupling the voltage-sensor activation to conformational changes in the selectivity filter of the pore module. By identifying the hitherto missing link between these well-documented structural changes, this study sheds new light on the fundamental mechanisms of electromechanical coupling.

**Decision letter after peer review:**

Thank you for submitting your article "Molecular basis for noncanonical coupling of the voltage sensor and the selectivity filter in *Shaker* K^+^ channels" for consideration by *eLife*. Your article has been reviewed by three peer reviewers, and the evaluation has been overseen by a Reviewing Editor and Richard Aldrich as the Senior Editor. The following individual involved in review of your submission has agreed to reveal their identity: John B Cowgill (Reviewer #2).

The reviewers have discussed the reviews with one another and the Reviewing Editor has drafted this decision to help you prepare a revised submission.

Summary:

The manuscript by Bassetto et al. continues work from the Bezanilla laboratory to identify the molecular basis of crosstalk between VSD and SF gates in *Shaker* K channels. Previous work from the group (and some data presented here) show that the voltage-sensor movement affects c-type inactivation which is primarily mediated by residues that constitute the selectivity filter in *Shaker* potassium channel. Careful measurements of the charge-voltage curves and conductance voltage curves, which are readouts for the voltage-sensor movement and pore gating, led them to identify a network of residues between the selectivity filter and voltage-sensor that mediate this long range interaction. By identifying a new pathway for voltage-sensor-selectivity filter coupling, the study makes important new contribution to the emerging body of literature showing that the VSD interactions with pore domain are more diverse than previously anticipated. Nonetheless, the reviewers noted few major concerns that should be addressed in the revised version.

Essential revisions:

1) A general concern for all the reviewers is whether the interacting network identified in this study is also important in the wild type channels since most of the scanning mutations were done in the mutant background L366R-W434F. While the motivation for choosing this double mutant background is evident, it is necessary to establish that the observed connectivity is not a mutant specific. One possible way is to show the energetic linkage by using another non-conducting mutant such P475A or other bundle crossing mutants that Kenton Swartz's group had identified. These Q-V measurements can be combined using thermodynamic mutant cycle (as was done by Fernandez et al., 2018) to show that the identified residues in the network are energetically connected. Moreover, these measurements will provide a quantitative assessment of the strength of these interactions.

2) The other concern was that the network identification was based on G-V/Q-V crossover assay. In the context of the manuscript, the mechanism underlying the crossover does not seem to be explicitly defined except in the context of a two-state scheme that is oversimplified, and actually does not account for allosteric coupling (i.e. distinct VSD activation steps that are allosterically coupled to a pore-opening step). It seems that the crossover might (alternatively) be explained by a direct effect on the kinetics of VSD activation or deactivation, or on the effective gating valence for VSD activation, rather than an effect on canonical coupling. These possibilities should be discussed or at least considered. But the experiments proposed in point #1 will address these concerns.

3) The reviewers also found the terminology referring to canonical and non canonical somewhat confusing. Some of the comments are appended:

"Discussion paragraph five is a little confusing. The discussion of KcsA seems tangential at first before being related to the *Shaker* channel. Also, the original canonical coupling definition in the Introduction states that canonical coupling is between the S4-S5 linker and inner gate, whereas the Discussion describes canonical coupling as between the inner gate and SF. This paragraph could be clarified by discussing the VSD-inner gate-SF coupling as indirect and the non-canonical coupling pathway as direct."

"The "coupling of the voltage sensor and the selectivity filter" in the title of the manuscript and in text may not be precise. Is the selectivity filter or the activation or inactivation gate coupled to the VSD?"

It will be helpful to use a more specific term to refer to this coupling between the voltage-sensor and C-type inactivation. The term non canonical coupling has been previously used to describe coupling between voltage-sensor and activation gates that are mediated via S5 and used in this context, it can be confusing. It will be better that the authors use a more specific language such as "C-type pathway" or "voltage-sensor selectivity filter coupling".

4) In Figure 1, the positions of L361 and L366 (in the voltage sensor) are shown relative to the newly identified chain of residues in the pore connecting the VSD to the selectivity filter. It is clear how L366 would interact with this chain of residues, but L361 appears >10 angstroms away. Presumably the interaction of L361 with these pore residues would occur in the resting state of the channel as the authors proposed in a previous work. However, this is not discussed anywhere in this paper and I think it would be helpful for a wide audience to see a comparison of the position of these residues in a resting state model (Jensen et al., 2012, for example).

---

## [Author Response]

Essential revisions:1) A general concern for all the reviewers is whether the interacting network identified in this study is also important in the wild type channels since most of the scanning mutations were done in the mutant background L366R-W434F. While the motivation for choosing this double mutant background is evident, it is necessary to establish that the observed connectivity is not a mutant specific. One possible way is to show the energetic linkage by using another non-conducting mutant such P475A or other bundle crossing mutants that Kenton Swartz's group had identified. These Q-V measurements can be combined using thermodynamic mutant cycle (as was done by Fernandez et al., 2018) to show that the identified residues in the network are energetically connected. Moreover, these measurements will provide a quantitative assessment of the strength of these interactions.

We understand the reviewers’ concerns about whether the interacting networking chain is also present in WT channels. In order to resolve the matter, we used the mutation V478W to record gating currents. This mutation, identified by Kenton Swartz’s group, is in the bundle crossing region. In V478W background, we studied S411 and F433 residues by thermodynamic mutant cycle analysis following what Fernandez et al., 2018 have published, as suggested. We took two residues of our proposed amino acids chain as a general case for practicality, avoiding the massive amount of experimental work if we were to test all possible pairs of residues. The outcome of our analysis demonstrated that all the residues in the chain modify the median of the Q-V curves and that residues S411 and F433 are indeed interacting (+2.00 kcal/mol). Therefore, the data shows the generality of the proposed chain that influences the VSD even without W434F, as well as an interaction between two members of the chain.

2) The other concern was that the network identification was based on G-V/Q-V crossover assay. In the context of the manuscript, the mechanism underlying the crossover does not seem to be explicitly defined except in the context of a two-state scheme that is oversimplified, and actually does not account for allosteric coupling (i.e. distinct VSD activation steps that are allosterically coupled to a pore-opening step). It seems that the crossover might (alternatively) be explained by a direct effect on the kinetics of VSD activation or deactivation, or on the effective gating valence for VSD activation, rather than an effect on canonical coupling. These possibilities should be discussed or at least considered. But the experiments proposed in point #1 will address these concerns.

Yes, in fact we cannot rule out the possibility that the W434F mutation is changing the valence and/or affecting the kinetics of the voltage sensor. We considered the suggestion in the manuscript, updating the text as follow:

”Another possibility is that the W434F destabilizes the resting state of L361R VSD channel, affecting the VSD activation and/or deactivation, seen as a right shift in the Q-V curve”.

However, we did not further expand it because, as the reviewer says, the experiments done in response to the previous comment address this concern.

3) The reviewers also found the terminology referring to canonical and non canonical somewhat confusing. Some of the comments are appended:"Discussion paragraph five is a little confusing. The discussion of KcsA seems tangential at first before being related to the Shaker channel. Also, the original canonical coupling definition in the Introduction states that canonical coupling is between the S4-S5 linker and inner gate, whereas the Discussion describes canonical coupling as between the inner gate and SF. This paragraph could be clarified by discussing the VSD-inner gate-SF coupling as indirect and the non-canonical coupling pathway as direct.""The "coupling of the voltage sensor and the selectivity filter" in the title of the manuscript and in text may not be precise. Is the selectivity filter or the activation or inactivation gate coupled to the VSD?"It will be helpful to use a more specific term to refer to this coupling between the voltage-sensor and C-type inactivation. The term non canonical coupling has been previously used to describe coupling between voltage-sensor and activation gates that are mediated via S5 and used in this context, it can be confusing. It will be better that the authors use a more specific language such as "C-type pathway" or "voltage-sensor selectivity filter coupling".

We appreciate the reviewers for the opportunity to further clarify this important topic in our study. We rearranged the paragraph highlighting the relevance of the two pathways that the voltage sensor couples with inactivation as follow:

”In *Shaker* channel, the residue I470 is a key participant in the coupling between the activation gate, at the bundle crossing region, and C-type inactivation (Peters et al., 2013). It has been proposed that I470 interacts with T442, in the signature sequence, TVGYG, at the selectivity filter (Labro et al., 2018). This mechanism constitutes an indirect pathway that couples the voltage sensor with the inner gate and the selectivity filter. Several studies have shown that mutations in the outer mouth of the pore, S5 segment and P-loop, affect C-type inactivation, highlighting the importance of these motifs for the inactivation process (De Biasi et al., 1993; Larsson and Elinder, 2000; Molina et al., 1997; Ortega-Saenz et al., 2000; Perozo et al., 1993; Yang et al., 1997). These studies and our data demonstrate the existence of a direct interaction between the voltage sensor and the selectivity filter mediated by the mechanism proposed here”.

We also standardized what was referred here as noncanonical as voltage-sensor selectivity filter gate coupling and we used the acronym VS-SF coupling.

4) In Figure 1, the positions of L361 and L366 (in the voltage sensor) are shown relative to the newly identified chain of residues in the pore connecting the VSD to the selectivity filter. It is clear how L366 would interact with this chain of residues, but L361 appears >10 angstroms away. Presumably the interaction of L361 with these pore residues would occur in the resting state of the channel as the authors proposed in a previous work. However, this is not discussed anywhere in this paper and I think it would be helpful for a wide audience to see a comparison of the position of these residues in a resting state model (Jensen et al., 2012, for example).

We appreciate the reviewers for raising the matter. We updated the Figure 1 showing the VSD in the resting state, highlighting residues L361 and L366, using the consensus model (Vargas et al., 2011, 2012)